# Smart-seq2 Technology Reveals a Novel Mechanism That Zearalenone Inhibits the In Vitro Maturation of Ovine Oocytes by Influencing *TNFAIP6* Expression

**DOI:** 10.3390/toxins15100617

**Published:** 2023-10-17

**Authors:** Zongshuai Li, Yali Liu, Tian Ma, Chen Lv, Yina Li, Hongwei Duan, Xingxu Zhao, Jianlin Wang, Yong Zhang

**Affiliations:** 1State Key Laboratory of Grassland Agro–Ecosystems, Key Laboratory of Grassland Livestock Industry Innovation, Ministry of Agriculture and Rural Affairs, Grassland Agriculture Engineering Center, Ministry of Education, College of Pastoral Agriculture Science and Technology, Lanzhou University, Lanzhou 730020, China; lizs@lzu.edu.cn; 2Gansu Key Laboratory of Animal Generational Physiology and Reproductive Regulation, Gansu Agricultural University, Lanzhou 730070, China; mt13893610637@163.com (T.M.); zoo_monkeys@163.com (C.L.); lyn9097@163.com (Y.L.); grand6138@163.com (H.D.); zhaoxx@gsau.edu.cn (X.Z.); 3Lanzhou University Second Hospital, Lanzhou 730030, China; wo_lyl@163.com

**Keywords:** Zearalenone, ovine oocyte, Smart-seq2, *TNFAIP6*

## Abstract

Zearalenone (ZEN), a non-steroidal estrogenic fungal toxin widely present in forage, food, and their ingredients, poses a serious threat to animal and human reproductive health. ZEN also threatens ovine, a major source of human food and breeding stock. However, the mechanisms underlying the impact of ZEN on the in vitro maturation (IVM) of ovine oocytes remain unclear. This study aimed to elucidate these mechanisms using the Smart-seq2 technology. A total of 146 differentially expressed genes were obtained, using Smart-seq2, from sheep oocytes cultured in vitro after ZEN treatment. ZEN treatment inhibited *RUNX2* and *SPP1* expression in the PI3K signaling pathway, leading to the downregulation of *THBS1* and ultimately the downregulation of *TNFAIP6*; ZEN can also decrease *TNFAIP6* by reducing *PTPRC* and *ITGAM*. Both inhibit in vitro maturation of ovine oocytes and proliferation of cumulus cells by downregulating *TNFAIP6*. These findings provide data and a theoretical basis for elucidating ZEN’s toxicity mechanisms, screening therapeutic drugs, and reducing ZEN-related losses in the ovine industry.

## 1. Introduction

The non-steroidal estrogenic fungal toxin zearalenone (ZEN), also known as F-2 toxin, is mainly produced by *Fusarium* species such as *F. graminearum* [1,2]. It was first isolated from moldy corn contaminated with *F. graminearum* and is one of nature’s most common fungal toxins [3]. ZEN (C_18_H_22_O_5_) has an estrogen-like structure; therefore, it has reproductive toxicity and primarily harms female animals [4,5]. ZEN can be produced at any point during crop development, maturity, harvest, or storage. It is found in various forage, food, and raw materials, including corn, rice, soybean, rye, and wheat, with corn having the highest detection rate and contents [6]. ZEN has a melting point of 161–163 °C, and it is not particularly sensitive to environmental changes or heat treatment. Therefore, it can stably exist during the storage and processing of feed, food, and byproducts [4,7]. Ovines are a primary source of human food and a major breeding stock in the livestock industry. Their coarse feed primarily comprises corn silage, corn straw, and alfalfa, all of which are sensitive to ZEN [5,7,8]. ZEN therefore poses a significant threat to ovine farming. ZEN can remain in meat products and enter the human body, and, as with other food contaminants, can pose a risk to human reproductive health [8].

In mice, ZEN can reduce the expression of cyclin B1 and CDK1, inducing G2/M phase arrest via the IPCHK1/2–Wee1–MPF pathway, and thus interfering with oocyte maturation [9]. Moreover, in mice, single-cell sequencing revealed that ZEN altered the status of germ cell and granulosa cells via the Hippo signaling pathway, and blocked the assembly of mouse primordial follicles [10]. In pigs, ZEN can significantly increase the phosphatidylcholine or phosphatidylethanolamine content and consume hemolysin phosphatidylcholine, thereby impairing follicular cavity formation and oocyte maturation [11]. In mice and pigs, ZEN can increase oxidative stress and disrupt spindle assembly and chromosome separation, leading to premature condensation of chromatin and interfering with the expression of genes related to oocyte activity, spindle assembly, redox potential, and apoptosis, thereby inhibiting the discharge of polar bodies and cumulus expansion and affecting oocyte maturation [12,13]; it also causes autophagy, apoptosis, and oxidative stress and disrupts embryo development by affecting the expression of epigenetic histones and organelle structure and function (mitochondria, endoplasmic reticulum, Golgi apparatus, and lysosomes), in turn affecting oocyte maturation and ovarian follicle formation [14,15,16]. It has a damaging effect on preantral follicles in ovine ovaries and participates in oocyte autophagy, although the specific mechanism of toxicity is unknown [17].

Smart-seq2 sequencing is particularly suitable for studying early embryonic cell populations, making it possible to sequence limited oocyte samples without requiring the ten-times more expensive single-cell sequencing technology [18]. However, there have been no reports to date on the mechanisms underlying the impacts of ZEN on the in vitro maturation (IVM) of sheep oocytes using Smart-seq2 technology.

In summary, the toxic effects of ZEN on both animal and human reproductive health remain unclear. This research on its mechanisms of influence on sheep oocyte IVM, via Smart-seq2, is expected to reveal new pathways and genes that will improve research into its reproductive toxicity mechanisms. Referring to the functions exercised by the core pathways or genes in Smart-seq2 (such as oxidative stress, autophagy, etc.), this research is also expected to screen antagonistic drugs for ZEN, as well as provide a theoretical basis and data for reducing or even eliminating its threat to reproductive health.

## 2. Results

### 2.1. ZEN Concentration Screening

The increase in yolk gap in the three experimental groups indicates that ZEN has a significant toxic effect on sheep oocytes (Figure 1A). Based on qPCR, *SOD1* and *CCNB1* expression was the lowest in the 20 μmol/L treatment (Figure 1B). Although *GPX1* did not show significant changes compared to the other groups, the 20 μmol/L treatment had the lowest value compared to the other two treatment groups (Figure 1B). *CAT* has a dose effect, but it has been significantly upregulated in the 20 μmol/L treatment (*p* < 0.05) (Figure 1B). *CDK1* and *SOD2* were both inhibited by ZEN, but their expression was highest in the 20 μmol/L treatment (Figure 1B). These results indicate that 20 μmol/L was an effective working concentration for ZEN.

### 2.2. Data Validation and Basic Omics Analysis

The number of oocytes used for sequencing is listed in Table 1. After obtaining omics data, *GPX1*, *CAT*, *BMP15*, *SOD1*, *CDC20*, *GDF9*, *CDK1*, and *CCNB1* were randomly selected for qRT-PCR (Figure 1C). The qPCR and transcriptome results were consistent, indicating that the transcriptome data were accurate and reliable (Figure 1D). Based on the correlation heat map (Figure 2A), the correlation coefficients between samples were ≥0.70. The violin plot of the sample shows that there is no significant degradation outlier in the sample, and the obtained data are valid and reliable (Figure 2B). Screening via edgeR (criteria: q Value ≤ 0.1 and multiple difference ≥ 1.2) identified 102 upregulated genes and 44 downregulated genes; these were selected for subsequent analysis (Figure 2C).

### 2.3. Enrichment Pathway Analyses

GO analysis was performed on the differentially expressed genes (DEGs). The 20 most enriched GO terms belonged mainly to biological processes and were related to intercellular interactions, cell development, and immune responses (Figure 2D). KEGG analysis of DEGs revealed that ZEN mainly affected signaling pathways related to human diseases and environmental information processes such as MicroRNAs in cancer, the PI3K-Akt and JAK-STAT signaling pathways, systemic lupus erythematosus, the TGF-beta signaling pathway, and extracellular matrix-receptor interactions (Figure 2E).

GSEA analysis was conducted on the sequencing results: the P450, autophagy, and cancer-related pathways were activated (Figure 3A), whereas signaling pathways related to primary immunodeficiency, autoimmune deficiency, and various diseases were suppressed (Figure 3B). *ITGAM* and *MHC2* were suppressed, indicating that ZEN inhibited the ovine oocyte immune system.

### 2.4. Key Gene Mining

Genes related to oocyte maturation, cumulus expansion factor, oxidative stress, apoptosis, autophagy, and spindle assembly detection points in the transcriptome data were extracted and visualized as histograms (Figure 4A,C–F). The expression of most genes related to oxidative stress, oocyte apoptosis, and autophagy was promoted, while that of most genes related to spindle, and all (tested) genes related to oocyte nuclear maturation, and cumulus expansion factor was suppressed. Fluorescence staining results showed that ZEN treatment promoted the production and accumulation of ROS in sheep oocytes cultured in vitro (Figure 4B).

Using String software (https://string-db.org/, accessed on 12 October 2023) to analyze the 146 DEGs, a network diagram of their protein–protein interactions (PPIs) was obtained, with ovine as the reference species. A total of 68 predicted proteins was obtained (Figure 5A). By labeling gene expression, we discovered the distribution of the downregulated and upregulated proteins (Figure 5A). Through Markov clustering in String, we obtained four sets with proteins number ≥ 4 (a gene set with a proteins number of 15 was suppressed) (Figure 5B).

Through a literature review, we classified 68 genes into five main categories: cancer (Figure 5C), oocyte development (Figure 5D), proliferation and aging (Figure 5E), oxidative stress (Figure 5F), and immunity (Figure 5G). Their respective heat maps indicated that except for a large number of cancer-related genes being activated, the other four categories of genes had varying degrees of activation and inhibition.

### 2.5. Diagram of the Mechanism of the Effects of ZEN on Ovine Oocyte IVM

We constructed a PPI network from transcriptome data, including DEG, estrogen receptors, and six related gene sets. The results showed that DEGs were most closely related to oocyte maturation, cumulus expansion factor, and cell apoptosis (Figure 6A,C). After removing the gene sets of oxidative stress, spindle assembly detection points, and autophagy from the network, we screened proteins with connection numbers ≥3 to obtain a closely related proteins set network (Figure 6C) and conduct a joint analysis with the directed network of DEG corresponding proteins (Figure 6B).

Finally, we inferred a conceptual diagram of the mechanism whereby ZEN affects ovine oocyte IVM (Figure 7).

## 3. Discussion

ZEN toxicity is known to be dose-dependent [12]. This was also confirmed early in the experiment; however, the maturation rates of the 20 and 30 μmol/L groups were very similar. qPCR detection showed that genes related to oxidative stress and cell cycle obstruction had peaks in the 20 μmol/L treatment group. We speculated that when the concentration of ZEN was 30 μmol/L, oocytes activated a limited correction mechanism for survival, so it was no longer conducive to research. Therefore, we selected the 20 μmol/L treatment group with a peak for research.

The samples are prepared and transported in batches. There are batch differences between the samples; however, the sample repeatability is strongly correlated. GO analysis revealed that cell adhesion, cell development, and immune-related terms were mainly affected; KEGG analysis revealed that the PI3K signaling pathway was at the core position; GSEA revealed that various immune mechanisms, such as primary immunity, were suppressed, primarily due to downregulation of *ITGAM* and *MHC*. These results indicate that ZEN treatment mainly affects ovine oocyte IVM by affecting cell development, cancer, and immunity; however, the specific mechanisms require further analysis.

Previous studies have shown that ZEN affects animal oocyte maturation mainly via the expression of genes involved in oocyte maturation, cumulus expansion factor, oxidative stress, cell apoptosis, autophagy, and spindle assembly checkpoint [9,10,11,12,13,14,15,16,17]. The overall transcriptome data analysis revealed that ZEN induced oxidative stress, promoted apoptosis and autophagy, and inhibited oocyte development and spindle assembly, thus impeding ovine oocyte IVM. This is consistent with existing research results; hence, we further classified DEG. The results showed that the genes are mainly enriched in the PI3K signaling pathway and have an impact on IVM of ovine oocytes through the following five aspects: (1) Oocyte maturation, the overexpression of genes such as *CCND2*, *PDGFC*, *IFRD1*, *H2AFZ*, *INHBA*, *CD200*, and *RPS26* indicated that both groups of oocytes had undergone IVM [19,20,21,22,23,24,25]. However, in the experimental group, the expression levels of *TNFAIP6* related to cumulus cell proliferation [13], *SPP1* related to in vitro development of oocytes [26], *RUNX2* related to follicular development [27], and *C1QB* related to embryonic development were downregulated [28], indicating that ZEN may inhibit the process of oocyte maturation by influencing these genes. (2) Overexpression of cancer genes, such as *FAM20C*, *PDGFRB*, *LAMB2*, *PSME2*, *CAD*, *ANXA3*, *MBNL1*, *EWSR1*, *RPL23A*, *NPNT*, and *SERPINH1*, is consistent with the reports that ZEN promotes cancer occurrence, which will lead to disruption of the oocyte metabolic system [29,30,31,32,33,34,35,36,37,38,39]. (3) Cell proliferation and aging, as well as the expression of genes such as *CCN1*, *A1CF*, *IL7R*, *DHRS3*, and *FBLN1*, indicated that ZEN promotes cell aging and apoptosis [40,41,42,43,44]. (4) Oxidative stress, overexpression of *CPE* and *NDRG1*, indicated the generation of ROS in cells [45,46]. Downregulation of *DAPP1* and *APOD* leads to a decrease in ROS production and antioxidant capacity [47,48], ultimately leading to an increase in ROS in oocytes. (5) Immunity, downregulation of *LCP1*, *LRRN3*, and *ERAP2* indicated that ZEN can cause damage to the immune system, leading to cell apoptosis [49,50,51].

Next, we further analyzed the relationship among estrogen receptor genes, enriched differential genes, and the six selected gene sets. We found that for IVM of ovine oocytes, the DEGs enriched in the ZEN treatment group were mainly related to estrogen receptors, cumulus expansion factors, oocyte maturation, and apoptosis-related genes. Although ZEN treatment caused oxidative stress, promoted autophagy, and interfered with spindle assembly, these processes are closely related to apoptosis-related genes but not to DEGs. The results of the directed network graph indicate that in ovines, ZEN–estrogen receptor binding inhibits the expression of *RUNX2* and *SPP1* in the PI3K signaling pathway, thereby downregulating *THBS1* and ultimately *TNFAIP6*, which in turn inhibits ovine oocyte IVM and cumulus cell proliferation. Furthermore, reducing *PTPRC* and *ITGAM* expression downregulates the expression of genes related to PI3K signaling, and ultimately *TNFAIP6*, thereby promoting apoptosis and inhibiting cumulus cell proliferation.

## 4. Conclusions

This study used Smart-seq2 single-cell transcriptome technology to elucidate the mechanism in which ZEN affects ovine oocyte IVM. We hypothesize two mechanisms for its action. First, inhibiting the expression of *RUNX2* and *SPP1* in the PI3K signaling pathway causes the downregulation of *THBS1*, and ultimately that of *TNFAIP6*, thereby inhibiting ovine oocyte IVM and cumulus cell proliferation. Second, reducing *PTPRC* and *ITGAM* expression downregulates the expression of genes related to PI3K signaling, and ultimately *TNFAIP6*, thereby promoting apoptosis and inhibiting cumulus cell proliferation. Further, these findings reveal that ZEN can cause autophagy-related gene overexpression and ROS accumulation in ovine oocytes, although further research is needed to clarify the related mechanisms. This study provides data and a theoretical basis for improving the study of the mechanisms of ZEN toxicity, screening therapeutic drugs, and reducing ZEN-related losses in the ovine industry.

## 5. Materials and Methods

### 5.1. In Vitro Culture of Ovine Oocytes

Washing solution (40 mL) was prepared using 39 mL M199 (Biological Industries, Kibbutz Beit Haemek, Israel), 1 mL HEPES (1 M) (Sigma Aldrich, St. Louis, MO, USA), and 160 mg of bovine serum albumin (BSA; Sigma Aldrich). Then, 10 mL of maturation solution was prepared using 10 mL M199 (Biological Industries), 10 µL follicle stimulating hormone (FSH, 5 μg/mL; Sigma Aldrich), 10 µL luteinizing hormone (LH, 5 μg/mL; Sigma Aldrich), 10 µL estradiol (1 μg/mL; Sigma Aldrich), 100 µL double antibody (10,000 units/mL Penicillin G Sodium Salt, 10 mg/mL Streptomycin Sulfate; Biological Industries), and 60 mg BSA (Sigma Aldrich). Both culture media were disinfected using the 0.22 μm filter.

The sheep ovaries were removed immediately after slaughter and placed in sterile phosphate-buffered saline (PBS) containing double antibody (final concentration 10.0 units/mL Penicillin G Sodium Salt, 100 µg/mL Streptomycin Sulfate) at 37 °C before being transported back to the laboratory within 2–3 h. The ovaries were placed in a sterile beaker, rinsed 2–3 times with PBS, and immersed in PBS containing double antibodies (final concentration 10.0 units/mL Penicillin G Sodium Salt, 100 µg/mL Streptomycin Sulfate). The beaker with ovaries was placed in a 37 °C water bath for later use. A 5 mL syringe was used to extract the follicular fluid, which was injected into a centrifuge tube containing 10 mL wash solution. At the same time, a 35 mm cell culture dish was prepared on an ultra-clean table, a 3 mm × 3 mm square was drawn on the underside of the dish, washing solution was added, and the dish was placed in the incubator for later use.

The centrifuge tube containing the follicular fluid was rested for 5 min, disinfected, and placed on an ultra-clean table. The lower layer of liquid was gently removed and transferred into a spare 35 mm culture dish. Under a microscope, cumulus–oocyte complexes (COCs) were extracted and placed in a new 35 mm culture dish. The COCs were then moved to another new 35 mm culture dish. Finally, the COCs were moved into a four-well plate (Thermo Fisher Scientific, Waltham, MA, USA) with droplets of maturation solution, and transferred to an incubator (5% CO2, 37 °C) for cultivation. The criteria for selecting COCs were as follows: oocyte cytoplasm uniformly filled within the zona pellucida, with three or more layers of dense cumulus cells (Grade A), and one to three layers of cumulus cells (Grade B) surrounding the COCs.

### 5.2. Screening of ZEN Working Concentration

The control and experimental groups were designated as CK (no ZEN) and T1, T2, and T3 (10, 20, and 30 μmol/L, respectively, in the maturation solution). After 26 h, hyaluronidase was used to remove granulosa cells. The naked eggs were transferred into a centrifuge tube containing PBS, centrifuged at 106 g for 5 min, and the supernatant was discarded. Next, 350 µL of TRIzol reagent (Thermo Fisher Scientific, Waltham, MA, USA) was added, total RNA was extracted from cells according to the manufacturer’s protocol, and the final sample was stored at −80 °C. cDNA was prepared using the RevertAid First Strand cDNA Synthesis Kit (Thermo Fisher Scientific, Waltham, MA, USA) following the manufacturer’s protocol. A QuantiNova SYBR Green PCR Kit (Qiagen, Dusseldorf, Germany) was used for quantitative real-time PCR (qRT-PCR) assays.

ZEN can cause oxidative stress and cell cycle obstruction in oocytes. Therefore, qPCR was performed to quantify *SOD1*, *SOD2*, *CAT*, *GPX*, *CDK1*, and *CCNB1* expression (Table 1). Working concentrations were determined based on the expression of the six genes at various concentrations.

### 5.3. ZEN Sample Preparation and Delivery

We established control and experimental groups using the optimal working ZEN concentration for the experimental groups. Because a limited number of oocytes was obtained each time, the samples were prepared six times, with the control and experimental groups set up each time, and any two sets of replicated samples were randomly combined. Finally, the control and experimental groups each had three biological replicates, for a total of six samples (Table 2). The cell samples were sent to Genedenovo Biotechnology Co., Ltd. (Guangzhou, China) on dry ice to establish a cDNA library.

### 5.4. Immunofluorescence

The reactive oxygen specifications (ROS) detection assay kit (BioVision, San Francisco Bay, CA, USA) was used for detection. Cumulus cells were digested, and the naked eggs were rinsed twice with PBS before being collected in centrifuge tubes. A 100 μL reactive oxygen species (ROS) assay buffer was added, and the cells were gently blown and centrifuged at 106× *g* for 5 min. The supernatant was discarded and rinsed with PBS once. A 100 uL 1× ROS Label Buffer was added, and the resulting mixture was incubated at 37 °C in the dark for 45 min and centrifuged at 106× *g* for 5 min. The supernatant was discarded, and 100 μL ROS assay buffer was added. The cells were gently blown and centrifuged at 106× *g* for 5 min to discard the supernatant. A 100 uL ROS Inducer Buffer was added, the mixture was incubated in the dark for 1 h, and the results were observed under an inverted fluorescence microscope.

### 5.5. Smart-seq2 Data Validation, Visualization, and Analysis

Eight genes (*GPX1*, *CAT*, *BMP15*, *SOD1*, *CDC20*, *GDF9*, *CDK1*, and *CCNB1*) were randomly selected to validate the sequencing results (Tabel 1). After verifying the sequencing results, the enriched Gene Ontology (GO) terms and Kyoto Encyclopedia of Genes and Genomes (KEGG) pathways were subjected to omics analysis, supplemented with Gene Set Enrichment Analysis (GSEA), in order to select the important signaling pathways and genes and to make preliminary inferences about the mechanisms of action of ZEN.

## Figures and Tables

**Figure 1 toxins-15-00617-f001:**
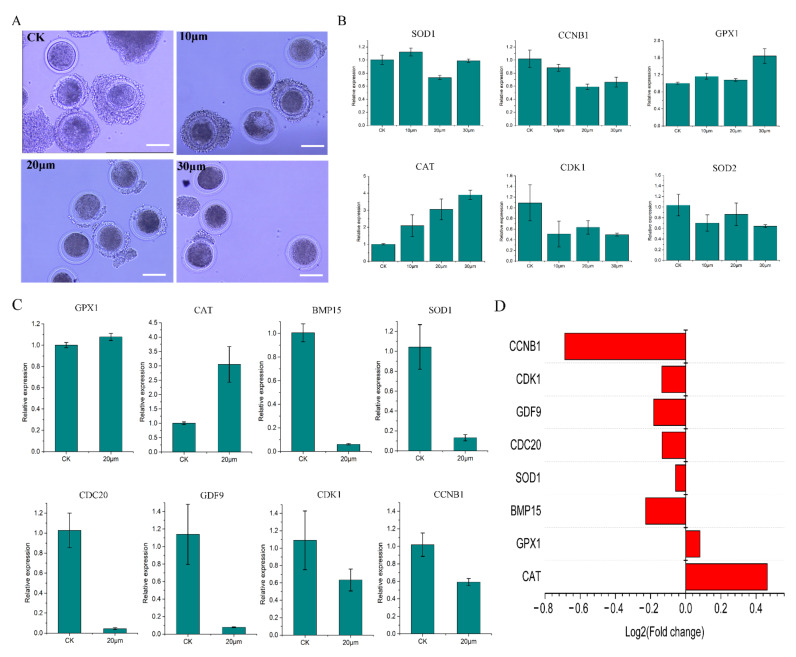
ZEN concentration screening and transcriptome data validation. (**A**) The growth status of sheep oocytes cultured in vitro after 36 h of exposure to different concentrations of ZEN(CK,10 μM, 20 μM, and 30 μM) (10×, 100 μm), (**B**) qRT-PCR results of *SOD1*, *CCNB1*, *GPX1*, *CAT*, *CDK1*, and *SOD2* in the four groups, (**C**) qRT-PCR results of randomly selected *GPX1*, *CAT*, *BMP15*, *SOD1*, *CDC20*, *GDF9*, *CDK1*, and *CCNB1*, (**D**) histogram of log2 values for *GPX1*, *CAT*, *BMP15*, *SOD1*, *CDC20*, *GDF9*, *CDK1*, and *CCNB* in transcriptome data.

**Figure 2 toxins-15-00617-f002:**
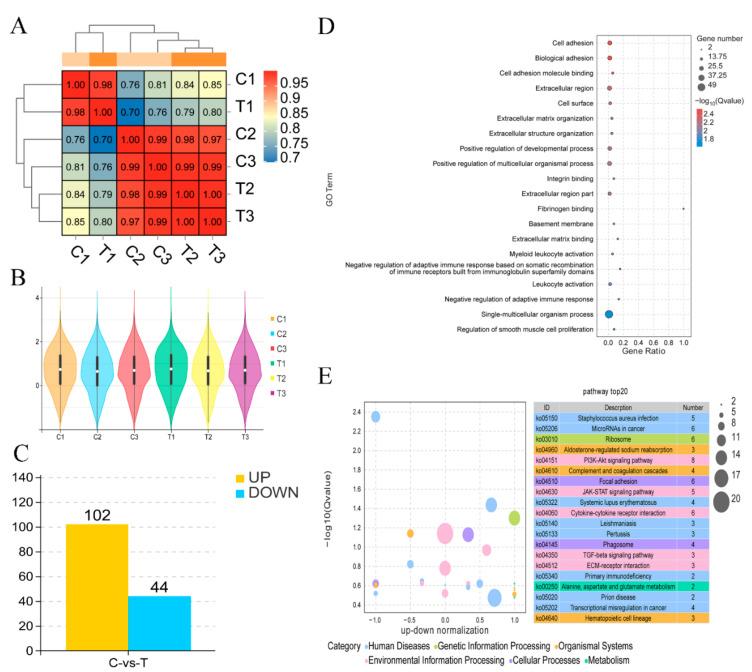
Visualization of transcriptome data. (**A**) Correlation heat map, (**B**) violin diagrams of six samples, (**C**) differentially expressed genes, (**D**) the 20 most enriched GO terms, and (**E**) KEGG analysis of DEGs.

**Figure 3 toxins-15-00617-f003:**
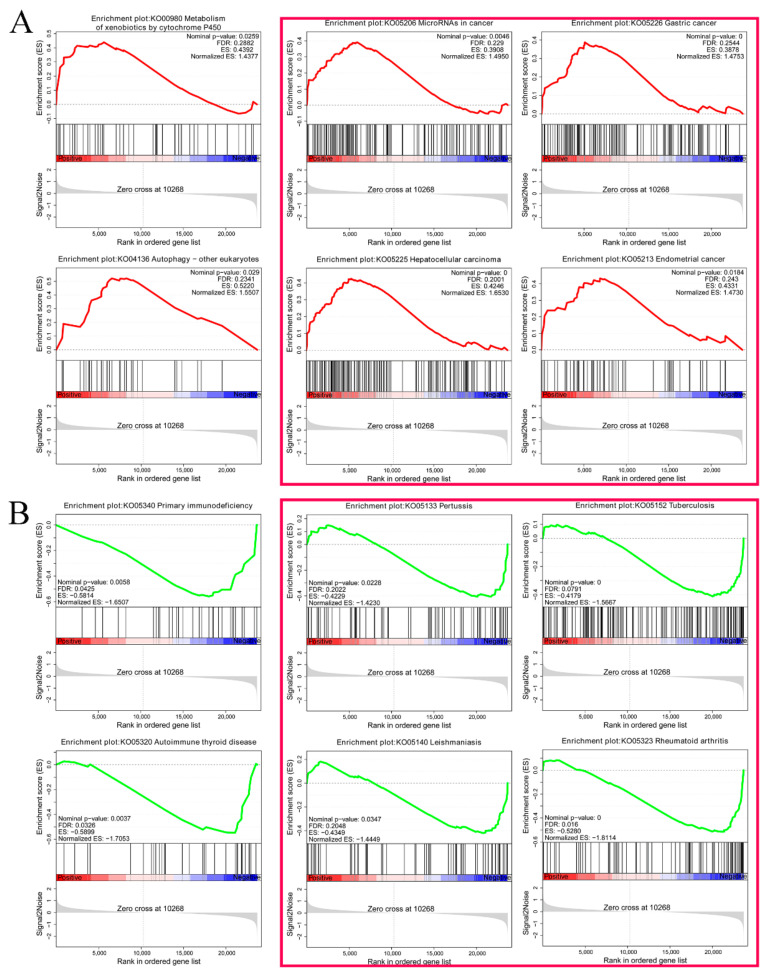
GSEA analysis. (**A**) Signal pathways upregulated by genes in GSEA analysis, and (**B**) signal pathways downregulated by genes in GSEA analysis.

**Figure 4 toxins-15-00617-f004:**
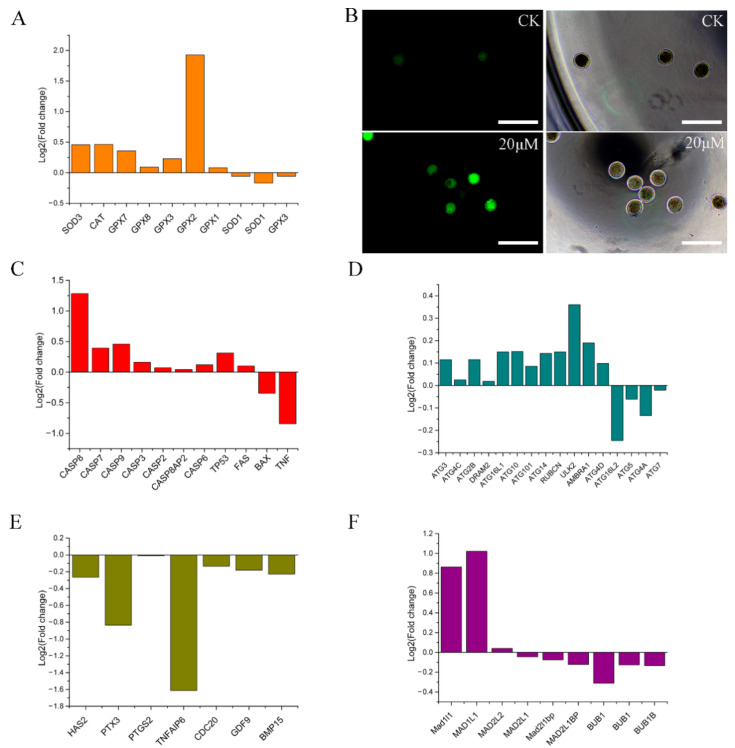
Visualization of transcriptome data. (**A**) Histogram of log2 values of oxidative stress related genes in transcriptome data, (**B**) fluorescence staining of reactive oxygen species in the control and experimental groups (fluorescent and white light) (20×, 50 μm), (**C**) histogram of log2 values of apoptosis related genes in transcriptome data, (**D**) histogram of log2 values of autophagy-related genes in transcriptome data, (**E**) histogram of log2 values of oocyte maturation and cumulus expansion factor related genes in transcriptome data, and (**F**) histogram of log2 values of spindle assembly detection point related genes in transcriptome data.

**Figure 5 toxins-15-00617-f005:**
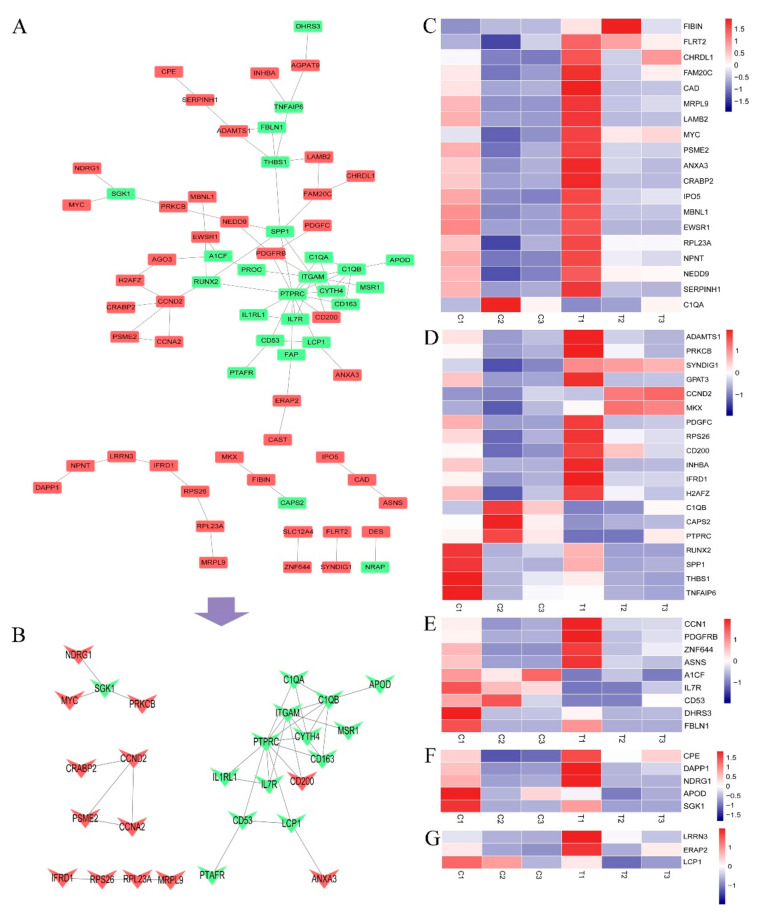
Gene classification analysis in DEGs. (**A**) PPI network diagram of DEGs, (**B**) four gene sets with gene number ≥4, (**C**) heat map of cancer related genes, (**D**) heat map of oocyte development related genes, (**E**) heat map of proliferation and aging related genes, (**F**) heat map of oxidative stress related genes, and (**G**) heat map of immunity related genes.

**Figure 6 toxins-15-00617-f006:**
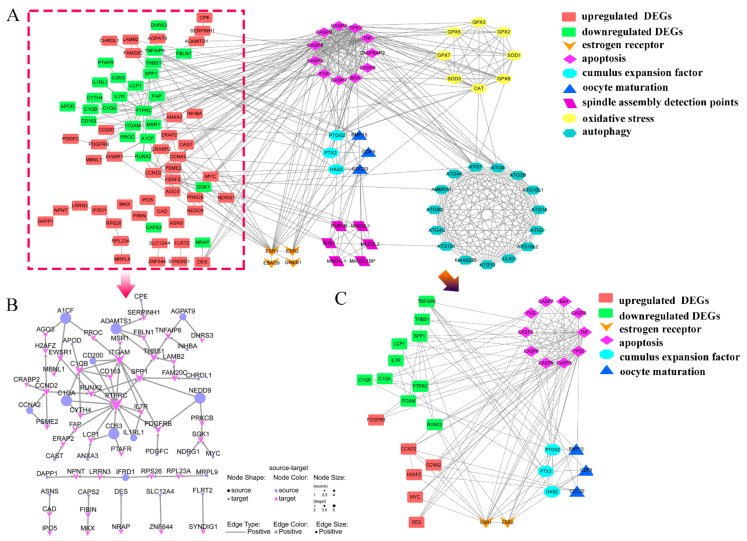
PPI analysis of DEGs. (**A**) PPI analysis of DEGs (within the red box) and its relationship with seven other selected genes, (**B**) the directed network graph of DEGs, (**C**) relationship diagram of proteins with connection number ≥3 in PPI.

**Figure 7 toxins-15-00617-f007:**
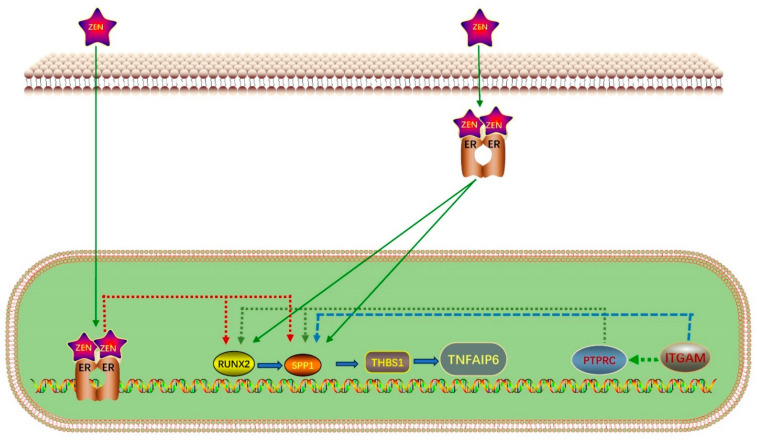
Conceptual diagram of the mechanism of ZEN on in vitro maturation of sheep oocytes.

**Table 1 toxins-15-00617-t001:** Primers for qRT-PCR.

Primers	Primer Sequences	Product Length (bp)
*SOD1*	GGCAATGTGAAGGCTGACAA	130
	TGCCCAAGTCATCTGGTCTT	
*SOD2*	GGACAAATCTGAGCCCCAAC	180
	CAATCTGTAAGCGTCCCTGC	
*CAT*	CCAGCCCTGACAAAATGCTT	242
	AAAGCGGGTCCTATGTTCCA	
*GPX*	CAGTTTGGGCATCAGGAAAAC	100
	CGAAGAGCATGAAATTGGGC	
*CDK1*	ATGGCTTGGATCTGCTCTCGAA	154
	TGCTCTTGACACAACACAGGA	
*CCNB1*	GCTTGGAGACATCGGTAACA	129
	GGAGCCTTTTCCAGAGGTTTTG	
*BMP15*	GGACACCCTAGGGAAAACCG	101
	TGTATGTGCCAGGAGCCTCT	
*CDC20*	GGCTGAGCTGAAAGGTCACA	214
	AACACCGTGAGGAGTTGGTC	
*GDF9*	TGACAGAGCTTTGCGCTACA	166
	TGATGGAAAGGTTCCTGCCG	

**Table 2 toxins-15-00617-t002:** The number of oocytes in Smart-seq2 sequencing samples.

Group	CK1	CK2	CK3	T1	T2	T3
Number of oocytes (pcs)	251	238	222	223	235	217

## Data Availability

The data that support the findings of this study are available from the corresponding author on reasonable request.

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
