# Peer review of "Smart-seq2 Technology Reveals a Novel Mechanism That Zearalenone Inhibits the In Vitro Maturation of Ovine Oocytes by Influencing *TNFAIP6* Expression"

_toxins, 2023, doi:10.3390/toxins15100617_

Round 1
Reviewer 1 Report
The authors report important data about ZEN-induced gene expression changes in oocytes. The manuscript requires some changes/clarifications before its acceptance for publication.
Introduction
Lines 31-32, and alfalfa, all of which are sensitive …
Line 33, body, and, as with …, can pose …
Line 39, the phosphatidylcholin
Lines 58-62, this is a rather unusual text formulation for the last paragraph of an introduction
Materials and methods
Line 67, HEPES?
Line 68, albumin
Line 74, 100 mL each?
Line 83, how disinfected?
Line 97 and several more times, please indicate the centrifugation speed in g
Lines 98 and 101, city and country of supplier is missing
Line 105, what about the 3 other genes in Table 1 (BMP15, CDC20, GDF9)?
Lines 119-120, this is not yet a full sentence
Line 123, and the cells were rinsed?
Line 130, there are 9 genes listed in Table 1
Line 136, 3. Results
Figure 1B, CAT, can the “positive” effect be explained?
Figure 1B, GPX1, is the effect of 20 mM ZEN compared to the control really significant?
Figures 1C and 1D do not fit; for instance, genes CDC20, GDF9 and BMP15 have about the same expression reduction according to Fig 1C, but this looks different in 1D.
Line 154, which gene pairs?
Figures 2F and 2G, the resolution is rather poor so that the text (enlarged in the electronic copy) is difficult to decipher
Line 184, of most genes,
Line 186, of most genes related to spindle, and all (tested) genes related to oocyte nuclear maturation, …
Lien 226, we inferred a conceptual diagram
Line 230, 4. Discussion
Line 280, PTPIC in Figure 6
Lines 296 and following, the references are not yet in the requested formatting style of the journal: abbreviated journal names shall be used; the punctuation is not yet correct, Volume to indicated but not issue; page range numbers.
Only minor changes required, e.g. missing articles (the).
Author Response
Response to Reviewer 1 Comments
Introduction
Point 1: Lines 31-32, and alfalfa, all of which are sensitive …
Response 1: Thank you for your question. I have modified " and alfalfa, and is sensitive to ZEN " to " and alfalfa, all of which are sensitive to ZEN ".
Point 2: Line 33, body, and, as with …, can pose …
Response 2: Thank you for your question. I have modified " body, as with other food contaminants, posing a risk to human reproductive health. " to " body, and, as with other food contaminants, can posing a risk to human reproductive health. ".
Point 3: Line 39, the phosphatidylcholin
Response 3: Thank you for your question. I have modified "phosphatidylcholin" to "the phosphatidylcholin ".
Point 4: Lines 58-62, this is a rather unusual text formulation for the last paragraph of an introduction
Response 4: Thank you for your question. I have made modifications to the wording in the article.
Materials and methods
Point 5: Line 67, HEPES?
Response 5: Thank you for your question. I have modified " Hepes " to " HEPES ".
Point 6: Line 68, albumin
Respons 6: Thank you for your question. I have modified " albumen " to " albumin ".
Point 7: Line 74, 100 mL each?
Response 7: Thank you for your question. I have made modifications to the wording in the article.
Point 8: Line 83, how disinfected?
Response 8: Thank you for your question. I added the disinfection method of the culture medium at the end of the first paragraph of the "2.1. In vitro culture of ovine oocytes" section.
Point 9: Line 97 and several more times, please indicate the centrifugation speed in g
Respons 9: Thank you for your question. I have made the modifications as required.
Point 10: Lines 98 and 101, city and country of supplier is missing
Respons 10: Thank you for your question. I have made the modifications as required.
Point 11: Line 105, what about the 3 other genes in Table 1 (BMP15, CDC20, GDF9)?
Respons 11: Thank you for your question. This table is a summary of the primers used in sections 2.2 and 2.5, with the remaining three genes being the required primers for section 2.5.
Point 12: Lines 119-120, this is not yet a full sentence
Response 12: Thank you for your question. I have made modifications to the wording in the article.
Point 13: Line 123, and the cells were rinsed?
Response 13: Thank you for your question. I have added relevant content.
Point 14: Line 130, there are 9 genes listed in Table 1
Response 14: Thank you for your question. I have added the specific names of eight genes.
Point 15: Line 136, 3. Results
Respons 15: Thank you for your question. I have modified " Discussion " to " Results ".
Point 16: Figure 1B, CAT, can the “positive” effect be explained?
Respons 16: Thank you for your question. Based on the results of reactive oxygen species detection, we believe that the upregulation of this gene is aimed at removing a large amount of reactive oxygen species generated in cells, which indirectly proves that the 20μmol/L treatment group has caused significant damage to cells. This is also one of the reasons why we chose 20μmol/L as the working concentration.
Point 17: Figure 1B, GPX1, is the effect of 20 mM ZEN compared to the control really significant?
Respons 17: Thank you for your question. Although GPX1 did not show significant changes compared to the other groups, the 20 μmol/L treatment group had the lowest value compared to the other two treatment group. So I also used this gene as one of the evidence to determine the working concentration. Considering the rigor of the article, I have clarified the corresponding content.
Point 18: Figures 1C and 1D do not fit; for instance, genes CDC20, GDF9 and BMP15 have about the same expression reduction according to Fig 1C, but this looks different in 1D.
Respons 18: Thank you for your question. The samples used for validation and sequencing were not prepared from the same batch. Due to the batch effect of samples prepared in different batches, we can only ensure that the trend of gene changes between the treatment group and the control group is consistent (upregulation or downregulation), but we cannot guarantee that the amount of gene changes in each batch of samples is exactly the same.
Point 19: Line 154, which gene pairs?
Response 19: Thank you for your question. I have made modifications to the wording in the article.
Point 20: Figures 2F and 2G, the resolution is rather poor so that the text (enlarged in the electronic copy) is difficult to decipher
Response 20: Thank you for your question. I have rearranged Figure 2 as required.
Point 21: Line 184, of most genes,
Respons 21: Thank you for your question. I have modified " of genes " to " of most genes ".
Point 22: Line 186, of most genes related to spindle, and all (tested) genes related to oocyte nuclear maturation, …
Respons 22: Thank you for your question. I have made the modifications as required.
Point 23: Lien 226, we inferred a conceptual diagram
Respons 23: Thank you for your question. I have modified " we inferred a network diagram " to " we inferred a conceptual diagram ".
Point 24: Line 230, 4. Discussion
Respons 24: Thank you for your question. I have modified " Conclusions " to " Discussion ".
Point 25: Line 280, PTPIC in Figure 6
Respons 25: Thank you for your question. I have modified " PTPIC " to " PTPRC " in Figure 6.
Point 26: Lines 296 and following, the references are not yet in the requested formatting style of the journal: abbreviated journal names shall be used; the punctuation is not yet correct, Volume to indicated but not issue; page range numbers.
Respons 26: Thank you for your question. I have made revisions to the references as required.

Reviewer 2 Report
I have received the Manuscript entitled: ‘Smart-seq2 technology reveals a novel mechanism that zearalenone inhibits the in vitro maturation of ovine oocytes by influencing TNFAIP6 expression‘ (Manuscript ID: toxins-2655653) submitted to the Toxins for a review.
Indeed, zearalenone is a non-steroidal estrogenic fungal toxin widely present in forage, food, and their ingredients, which poses a serious threat to animal and human health. Due to estrogen-like structure, it has reproductive toxicity and primarily harms female animals. Fluorescence staining results showed that zearalenone treatment promoted the production and accumulation of ROS in sheep oocytes cultured in vitro. These findings provide data and a theoretical basis for elucidating zearalenone’s toxicity mechanisms, screening therapeutic drugs.
The Manuscript is written with care and contains substantial elements of novelty. The methodological part is very well developed by the authors as well. They collected interesting and solid set of data. The significance of the gathered results is quite well presented, therefore, I consider the submitted Manuscript as a good candidate for publication in Toxins. However, there are some elements that require improvement by authors to facilitate clarity of the content and increase the potential interest of future readers:
· Line 60: Introduction - it can be beneficial to mention in 1-3 sentences how the revealed genes can be utilized in terms of drug discovery in general
· Line 176: authors should reconsider data presentation in this figure. Graphs from D to G are very difficult or impossible to read and the font is to small
· Line 209: Fig. 4A is too small as well
· Line 222: Fig. 5 - font too small quality to low in all points
· Line 230 and 283 - two conclusions sections are unnecessary. I recommend to remove the latter one or merge them together
Author Response
Response to Reviewer 2 Comments
Point 1: Line 60: Introduction - it can be beneficial to mention in 1-3 sentences how the revealed genes can be utilized in terms of drug discovery in general
Response 1: Thank you for your question. I have revised the original article to “Referring to the functions exercised by the core pathways or genes in Smart-seq2 (such as oxidative stress, autophagy, etc.), this research is also expected to screen antagonistic drugs for ZEN, as well as a theoretical basis and data for reducing or even eliminating its threat to reproductive health”.
Point 2: Line 176: authors should reconsider data presentation in this figure. Graphs from D to G are very difficult or impossible to read and the font is to small
Response 2: Thank you for your question. I have rearranged Figure 2 as required.
Point 3: Line 209: Fig. 4A is too small as well
Response 3: Thank you for your question. I have rearranged Figure 4 as required.
Point 4: Line 222: Fig. 5 - font too small quality to low in all points
Response 4: Thank you for your question. I have rearranged Figure 5 as required.
Point 5: Line 230 and 283 - two conclusions sections are unnecessary. I recommend to remove the latter one or merge them together
Response 5: Thank you for your question. I have modified " 4.Conclusion " at Line 230 to " 4.Discussion ".
